# Development and Evaluation of Different Electrospun Cysteamine-Loaded Nanofibrous Webs: A Promising Option for Treating a Rare Lysosomal Storage Disorder

**DOI:** 10.3390/pharmaceutics16081052

**Published:** 2024-08-09

**Authors:** Safaa Omer, Nándor Nagy, Balázs Pinke, László Mészáros, Adrienn Kazsoki, Romána Zelkó

**Affiliations:** 1Center of Pharmacology and Drug Research & Development, University Pharmacy Department of Pharmacy Administration, Semmelweis University, Hőgyes Endre Street 7-9, H-1092 Budapest, Hungary; safaa.omer@phd.semmelweis.hu; 2Department of Anatomy, Histology and Embryology, Semmelweis University, Tűzoltó Street 58, H-1094 Budapest, Hungary; nagy.nandor@semmelweis.hu; 3Department of Polymer Engineering, Faculty of Mechanical Engineering, Budapest University of Technology and Economics, Műegyetem Rkp. 3, H-1111 Budapest, Hungary; pinke@pt.bme.hu (B.P.);

**Keywords:** accelerated stability test, cysteamine-containing formulation, cytocompatibility study, electrospinning, in vitro drug release study, ocular insert

## Abstract

Nanofibers can be utilized to overcome the challenges faced by conventional ophthalmic formulations. This study aimed to develop and characterize cysteamine (Cys)-loaded nanofiber-based ophthalmic inserts (OIs) as a potential candidate for the treatment of ophthalmic cystinosis using water-soluble polyvinyl alcohol (PVA)/poloxamer 407 (PO-407) and water-insoluble tetraethoxysilane (TEOS)/PVA nanofibers. Plain and Cys-loaded fibers in different proportions were prepared by the electrospinning method and studied for their morphological, physicochemical, release study, cytocompatibility effects, and stability study. The fiber formation was confirmed by scanning electron microscopy, while Fourier transform infrared spectra showed the most critical peaks for the Cys and the excipients. The release of the Cys was fast from the two polymeric matrices (≤20 min). The release from TEOS/PVA nanofibers is characterized by Case II transport (0.75 < β < 1), while the release from PVA/PO-407 nanofibers follows Fickian diffusion (β < 0.75). The cytocompatibility of compositions was confirmed by hen eggs tested on the chorioallantoic membrane (HET-CAM) of chick embryos. All formulations remained stable under stress conditions (40 ± 2 °C, 75 ± 5% relative humidity) regarding morphology and physicochemical characteristics. The developed nanofibrous mats could be an excellent alternative to available Cys drops, with better stability and convenience of self-administration as OIs.

## 1. Introduction

Cystinosis is a rare autosomal recessive disorder resulting from mutations in the gene that encodes for the cystinosin protein (CTNS gene). Cystinosin is a protein responsible for the transportation of the amino acid cystine. A lack of this transporter leads to an abnormal accumulation of cystine crystals in various organs and tissues in the body, particularly affecting the kidneys and eyes, leading to harmful damage [1]. Three forms of cystinosis have been recognized: infantile nephropathic cystinosis, juvenile nephropathic cystinosis, and adult or ocular cystinosis with ocular cystinosis can be involved in all forms of the disease [2,3,4,5]. Ocular cystinosis is characterized by various symptoms, including photophobia, blepharospasm, and corneal erosion. Failure in treatment can lead to glaucoma, papilledema, vision impairment, or even blindness [6,7,8,9,10,11].

There are no generalized guidelines concerning ocular cystinosis treatment, but there are ongoing comprehensive studies involving the early diagnosis, treatment optimization, and prevention of disease prognosis [12]. The treatment modalities involve using cystine-depleting agents, mainly cysteamine (Cys), to prevent the systemic complications of cystinosis [13]. Due to the lack of corneal vascularization, orally administered Cys cannot reach the cornea. Therefore, systemic Cys is useless in treating ocular cystinosis [4,7,14]. Topical Cys formulations have been successfully used to control the level of cystine crystals in different eye layers, but eye drops have several limitations, including poor bioavailability, stability problems, frequent administration, precipitation of ocular irritation, and corneal epithelial erosion [13,15].

Alternative Cys formulations that aim to increase the residence time have been developed, including ophthalmic hydrogels, nano wafers, and contact lenses. These new formulations have a noticeable impact on the reduction of cystine crystals and blurred vision. Furthermore, they act as a lubricant and increase patients’ comfort and adherence because they are fabricated using the polymer biomaterials of artificial tear eye drops [16]. Drug carriers significantly enhance cystinosis treatment by improving the Cys permeation to target tissues, boosting stability, increasing residence time, facilitating controlled release, enhancing bioavailability, and reducing side effects [17,18].

Great efforts have been made to develop novel drug delivery systems as promising alternatives to conventional formulations for ophthalmic targeting [19,20,21]. These systems are designed to improve corneal residence time, ocular bioavailability, and comfort for the patients by overcoming the limitations of conventional eye drop formulations and lowering the complications arising from systemic and intravitreal administration [22,23,24]. These alternative formulations include ophthalmic gels, mini-tablets, biodegradable polymeric systems, collagen shields, implants liposomes, nanoparticles, implants, nanosuspensions, and ophthalmic inserts [17,18,25,26,27,28,29,30].

Ophthalmic inserts (OIs) have been intensively studied in recent years as innovative alternative drug delivery systems that aim to increase patients’ acceptability and bioavailability [31]. They are sterile formulations with a shape and size that enables their placement in the conjunctival sac [32,33]. OIs offer numerous advantages, such as increasing patient comfort by decreasing the ophthalmic irritation arising from preservatives. They are fabricated from different biocompatible polymers of various properties that aid in increasing the residence time, modulating the drug release, and increasing trans-corneal absorption and bioavailability. Moreover, they have a longer shelf life compared to liquid formulations [34,35,36]. OIs can be prepared by several methods, including solvent casting, melt extrusion, glass substrate technique, and electrospinning [37,38].

Electrospinning is a simple and versatile technique widely used in drug delivery applications [39]. The process utilizes electrostatic forces to produce nanofibrous films with unique properties. Electrospinning allows the flexible use of diverse polymers with tunable properties such as mucoadhesion and release modulation [40]. The process is suitable for the encapsulation of thermosensitive molecules as well as biological drug molecules. The method results in nanoscale formulations with a high surface-to-volume ratio that increases the ophthalmic drug contact and corneal absorption [34,38,41].

Various natural, semisynthetic, and synthetic polymers are used to prepare electrospun nanofibers. Synthetic polymers have superior quality over natural polymers because they offer several advantages, such as the flexibility for surface engineering, resistance to microbial contamination, tunable properties, and vulnerability for fabrication into different shapes, making them excellent candidates for tissue regeneration and biomedical applications [42]. In general, these polymers play a vital role in the determination of the final properties of the formulation. Some polymers provide a fast and complete release of active pharmaceutical ingredients (APIs), while other polymers can slow the release of APIs. Some of the polymers that are used individually or in combinations to prepare electrospun nanofiber-based OIs include polyglycolic acid, polylactic acid, polycaprolactone, polyvinyl alcohol (PVA), polyacrylic acid, polyvinyl pyrrolidone, hyaluronic acid, chitosan, polyethylene oxide, polymethacrylate, cellulose derivates, polyacrylamide, and carboxy vinyl polymers [34,35,43,44,45,46,47].

PVA is one of the polymers widely used as films and fibers forming polymers. It is a thermoplastic polymer, biodegradable, biocompatible, and possesses sufficient mechanical strength to be processed in different formulations. The electrospinning technique provides considerable benefits to produce PVA nanofibers, such as versatility, control over fiber properties, and ease of processing. Although alternative methods such as solvent casting, glass substrate, and melt extrusion are available, electrospinning remains the most widely used and effective technique for the fabrication of PVA nanofibers, with tailored characteristics for a wide range of applications. Therefore, it has been exploited in many areas, including biomedical and pharmaceutical applications [48,49,50,51].

The modification of PVA to impart additional properties can occur by crosslinking with glutaraldehyde, surface modification with the biopolymer, or heat treatment. Some studies manipulated the modification of PVA by the formation of PVA/silica nanocomposites, which can reduce the hydrophilicity of PVA by occupying the abundant hydroxyl groups in the PVA backbone through chemical interaction with silica (tetraethoxysilane, TEOS), therefore sustaining the release of hydrophilic drugs from hydrophilic polymers might be obtained by developing hydrogel systems [46,52,53]. This system involves the production of a PVA/silica hybrid with enhanced stability while maintaining the biocompatibility of the PVA [54].

A blend of polymers may be used to modulate the drug release, impart mucoadhesive properties, or change the solubility characteristics [35]. For example, adding poloxamer 407 (PO-407) to the ocular formulation to improve the API’s ocular delivery is very common [55]. PO-407, also known as Pluronic F127, is a common component in ophthalmic formulations due to its unique thermoreversible and amphiphilic behavior that enables micelle formation and the encapsulation of hydrophobic drugs within an aqueous environment. PO-407 plays a key role in solubilizing drugs and enhancing the adhesiveness of the formulations [56,57].

The current study aims to develop a novel, solid formulation for targeting orphan disease, ocular cystinosis, by formulation of Cys-loaded nanofiber-based OIs as an alternative to conventional Cys eye drops. The study compares the release of Cys from two polymeric blends, a water-soluble matrix (PVA/PO-407) and a water-insoluble matrix (TEOS/PVA).

## 2. Materials and Methods

### 2.1. Materials

Cysteamine hydrochloride (CysH) was obtained from Sigma-Aldrich Chemie GmbH (Schnelldorf, Germany). Ellman’s reagent ((5,5′-dithio-(bis-2-nitrobenzoic) acid, DTNB for UV-visible quantification of thiol compound, a product of TCI Ltd. (Tokyo, Japan)). Polyvinyl alcohol (PVA, Mowiol^®^ 8–88, Mowiol^®^ 18–88, and Mowiol^®^ 40–88 with an average molecular weight Mw~67 kDa, Mw~130 kDa, Mw~205 kDa), poloxamer 407 (PO-407, average molecular weight, Mw~12.6 kDa), hydrochloric acid, and tetraethoxysilane (TEOS) were obtained from Merck Ltd. (Budapest, Hungary). Polysorbate 80 (PS-80), potassium dihydrogen orthophosphate, sodium hydroxide, ethanol (anhydrous, 96%), and ethylenediaminetetraacetic acid (EDTA) were purchased from Molar Chemicals Ltd. (Budapest, Hungary). Pharmacopeial-grade distilled water was used as a solvent for precursor solution preparation.

### 2.2. Methods

#### 2.2.1. Preparation of Polyvinyl Alcohol (PVA)/Poloxamer 407 (PO-407) Precursor Solutions

The precursor solutions for pure PVA and PVA/PO-407 fibers were created using three grades of PVA and PO-407. The preparation of each solution involved specific quantities of polymer and distilled water, as detailed in Appendix A. To create the PVA solutions, the polymers were dispersed in distilled water and then stirred under heat until clear, viscous solutions formed. For PO-407 solutions, the required amount was dissolved in cold water. The PVA/PO-407 solutions were produced by combining precise amounts of each solution and stirring at room temperature for 2 h. Additional solutions containing PS-80 (0.5% (*w*/*w*) and 1% (*w*/*w*)) were also prepared following the proportions outlined in Appendix A.

#### 2.2.2. Preparation of Polyvinyl Alcohol (PVA)/Tetraethoxysilane (TEOS) Precursor Solutions

To prepare TEOS/PVA precursor solutions, a mixture of TEOS/EtOH/H_2_O/HCl in a molar ratio of 1:3:8:0.04 was initially prepared in a tightly sealed glass container and heated to 60 °C. After allowing 3 h for hydrolysis, solutions of PVA at concentrations of 10%, 12%, and 14% (*w*/*w*) in water/ethanol (8:2 mass ratio) were added in four different mass ratios (TEOS/PVA solutions at 1:4, 2:3, 3:2, and 4:1). This mixture was then stirred for one hour at 60 °C. The impact of varying PVA concentrations on the morphology of the fibers was assessed at three levels (10%, 12%, and 14% *w*/*w*). For each PVA concentration, four different TEOS/PVA mass ratios (4:1, 3:2, 2:3, and 1:4) were evaluated (see Appendix A).

#### 2.2.3. Preparation of Cysteamine (Cys)-Loaded Viscous Solutions

To prepare Cys-loaded solutions, 0.0055 g and 0.011 g of CysH were added to 1 g of plain PVA/PO-407 and TEOS/PVA viscous solutions to obtain the final solution concentration of 0.55% (*w*/*w*) and 1.1% (*w*/*w*). The drug-loaded solutions were stirred at room temperature for 2 h until complete homogenization (Table 1 and Table 2).

#### 2.2.4. Electrospinning of the Solutions

Fibers were prepared using laboratory-scale electrospinning equipment (SpinSplit Ltd., Budapest, Hungary). Each sample involved filling a 1 mL plastic syringe (Luer lock syringe, Merck Ltd., Budapest, Hungary) with either plain or drug-loaded viscous solutions. The syringe was then connected to a 22 G conventional needle via a Teflon tube and attached to a pump to regulate the solution flow. After numerous preliminary experiments, the process parameters were optimized, setting the electrospinning process at a flow rate of 0.08–0.1 µL/sec, an applied voltage of 10–20 kV, and effective distances between the needle and the grounded collector at 10, 12.5, and 15 cm. The samples were collected on aluminum foil wrapped around the grounded plate collector and stored in a desiccator until further analysis. The process was carried out under ambient conditions of 22 ± 1 °C room temperature and 40 ± 5% relative humidity.

#### 2.2.5. Morphological Characterization

The morphological analysis of the samples was conducted using a JEOL JSM-6380LA (Tokyo, Japan) scanning electron microscope (SEM). The samples were mounted on copper ingots using double-sided carbon adhesive and coated with gold under vacuum. Images were captured at 3500× and 5000× magnifications, with a working distance of 10 mm and an accelerating voltage of 10 kV. The images were evaluated for fibrous, non-fibrous, and bead-containing samples. For fibrous samples, diameter measurements were performed on 100 randomly selected individual fibers (*n* = 100) from two images using ImageJ v1.46r software. The average fiber diameters and standard deviations were calculated using Excel 2010. Histograms and Gaussian distribution fitting were carried out using OriginPro 2018 software (v9.5.1., OriginLab Corporation, Northampton, MA, USA). The normality of fiber diameter distribution, skewness, and kurtosis were calculated using Microsoft Excel 2010 functions according to the specified Equations (1) and (2):(1)$$Skew=\frac{n}{\left(n-1\right)\left(n-2\right)}\Sigma ({\frac{xi-\bar{x}}{s})}^{2}$$
(2)$$Curtosis=\frac{n}{\left(n-1\right)\left(n-2\right)\left(n-3\right)}\Sigma ({\frac{xi-\bar{x}}{s})}^{4}\}-\frac{3({n-1)}^{2}}{\left(n-2\right)\left(n-3\right)}$$
where *n* is the number of data points, *xi* is the mean, and *s* is the standard deviation.

#### 2.2.6. Fourier Transform Infrared Spectroscopy

Fourier transform infrared spectroscopy (FTIR) was employed to analyze the physicochemical characteristics, compatibility, and molecular interactions between polymers and other excipients. The analysis was conducted on individual components as well as their fibrous mixtures, including both plain and drug-loaded samples. Using a Jasco FT/IR-4200 spectrophotometer (Jasco Inc., Easton, MD, USA), measurements were taken at ambient temperature with the following parameters: a spectral range of 400–4000 cm^−1^, a resolution of 4 cm^−1^, and an average of 100 scans.

#### 2.2.7. Energy Dispersive X-ray Spectroscopy (EDAX)

Prior to analysis, the samples F11 and FT27 (PVA/PO-407 and TEOS/PVA blends, respectively) were coated with a thin layer of gold using a JEOL JFC-1200 spray coating machine. A Jeol JSM-6380LA scanning electron microscope (SEM) equipped with an energy dispersive X-ray spectroscopy (EDAX) detector was used for the examination and analysis of the samples. An accelerating voltage of 15 kV was used during the analysis to generate X-ray photons from the samples. The atomic composition of the samples was determined from the spectra and their distribution was determined by a mapping method.

#### 2.2.8. Determination of the Drug Content

To quantify the Cys content in each fibrous sample, the following procedure was employed:

Sample preparation:For PVA/PO-407-based matrices: Approximately 38 mg (equivalent to 10 μM) was weighed.For TEOS/PVA-based matrices: Approximately 75 mg (equivalent to 10 μM) was weighed.

Dissolution:The weighed sample was dissolved in 3 mL of phosphate buffer (pH 7.4).The solution was stirred at room temperature for 30 min.

Reaction and measurement:10 μL of Ellman’s reagent was added to the dissolved sample.The absorbance was measured at a wavelength of 412 nm.A Jasco 530 UV-VIS spectrophotometer with an inline probe was used for the measurement.

Data analysis:Measurements were performed in triplicate.Cys content was calculated using a pre-established calibration curve.

This method allows for the accurate determination of Cys content in different fibrous matrix compositions.

#### 2.2.9. In Vitro Release Study and Release Kinetics

The in vitro release of Cys was studied based on a modified method analogy to the basket method reported by Pharmacopoeia Hungarica (Ph.Hg. VIII) [58]. The technique was developed to hold small-volume dissolution media to simulate small physiological compartments such as the buccal cavity and ophthalmic sac. Samples containing 10 μM were weighed, folded in a dry magnetic bar, and inserted inside the 25 mL beaker. The beaker was placed on a magnetic stirrer adjusted to provide 100 rpm at a temperature of 35 ± 0.5 °C. An in-line probe of a Jasco-V-750 UV-VIS spectrophotometer was immersed in the beaker to detect the absorbance of the released Cys. A pre-warmed dissolution media (10 mL phosphate buffer pH 7.4 containing 30 μL of Ellman’s reagent at 35 °C) was added to the 25 mL beaker containing the sample, and the absorbances were measured at a predetermined time interval (5 s) at λ_max_ 412 nm. The measurements were performed in triplicate, and the release curve was constructed from the average values of the measurements. A Weibull model was used to evaluate the kinetics of Cys release according to Equation (3):(3)Mt=M∞1−ⅇ−t−t0βτd
where *M_t_* is the Cys release at (*t*) time, while *M_∞_* is the maximum amount of the released Cys. The parameters *t*_0_ and *τ_d_* are the lag and average dissolution times, respectively. The *β* parameter plays a crucial role in defining the shape of the release curve: When *β* = 1, it indicates first-order kinetics, *β* > 1 suggests a release pattern with a slow initial phase followed by acceleration, and *β* < 1 implies a rapid initial release that subsequently slows down [59,60].

#### 2.2.10. Hen’s Egg Test on Chorioallantoic Membrane (HET-CAM)

The HET-CAM (Hen’s Egg Test-Chorioallantoic Membrane) assay was employed to assess the potential for eye irritation caused by the newly developed cysteine-loaded nanofibers. This test evaluates the occurrence of hyperemia, hemorrhage, and coagulation when the chorioallantoic membrane (CAM) of 9-day-old chicken embryos is exposed to the formulations under investigation. The study utilized fertilized eggs from White Leghorn chickens (*Gallus gallus domesticus*), sourced from a commercial breeder (Prophyl-BIOVO Hungary Ltd., Mohacs, Hungary). The HET-CAM test is widely recognized as an effective alternative to in vivo eye irritation testing, offering a reliable method to evaluate the ocular safety of various substances, including pesticides and pharmaceutical formulations. This approach aligns with efforts to reduce animal testing while still providing valuable insights into potential ophthalmic irritation. The eggs were maintained at a temperature of 37.5 °C ± 0.5 °C in a humidified HEKA 1+ egg incubator (Rietberg, Germany). After the development of CAM on the 9th day of incubation, a small hole was made in the hard shell and expanded to about 2 cm with ophthalmic surgical scissors at the blunt end of the eggs. The inner membrane was carefully removed to expose the vascularized CAM. The plain and Cys-loaded nanofiber mats (38 mg of Cys-loaded and plain fibers for PVA/PO-407-based systems and 75 mg of Cys-loaded and plain fibers for TEOS/PVA-based systems) were placed on the surface of the vascularized CAM and evaluated against phosphate-buffered saline (PBS) pH 7.4 and 0.1 N NaOH solution as negative and positive controls, respectively. After 20 s of applying the tested materials, the vascular CAM was rinsed with 5 mL of PBS and evaluated for irritation effects. Images were captured at 0.5, 2, and 5 min using a Nikon SMZ25 stereomicroscope (Unicam Ltd., Budapest, Hungary). Image processing was conducted using Nikon’s proprietary software QCapture Pro (NIS-Elements Basic Research version obtained from AURO-SCIENCE Consulting Ltd., Budapest, Hungary). Each sample was given a score based on the numerical time-dependent scores for hyperemia, hemorrhage, and coagulation [61,62].

#### 2.2.11. Accelerated Stability Study

The accelerated stability study of Cys-loaded electrospun samples was conducted under controlled stressful conditions. The samples were collected in aluminum foils and sealed in hermetic zip-lock bags. They were then stored in a stability chamber (Sanyo type 022, Leicestershire, UK) for four weeks at 40 ± 2 °C and 75 ± 5% relative humidity. At predetermined intervals of 0, 1, 2, 3, and 4 weeks, the samples were evaluated for morphological and physicochemical changes using SEM and FTIR, respectively. These analyses were performed after subjecting the samples to elevated levels of temperature, humidity, and pressure to assess their stability under stress conditions.

#### 2.2.12. Statistical Analysis

OriginPro 2018 software (v9.5.1., OriginLab Corporation, Northampton, MA, USA) was used to construct figures and to carry out the statistical analysis. ANOVA tests were used to assess the differences between the data. A *p*-value of less than 0.05 was considered statistically significant. Microsoft Excel 2010 functions were applied to assess the normality of fiber diameter distribution, skewness, and kurtosis.

## 3. Results and Discussion

### 3.1. Morphological Characterization

#### 3.1.1. Polyvinyl Alcohol (PVA)/Poloxamer 407 (PO-407) Samples

The morphology of all samples was evaluated by SEM. The SEM images for morphological characterization of electrospun PVA/PO-407/PS-80 samples are displayed in Appendix A. PS-80 in a concentration of 0.5–1% (*w*/*w*) acts as nonionic surface-active agent that slightly liquifies the highly viscous PVA/PO-407 precursor solutions [63]. The final morphology of PVA/PO-407 blends with PS-80 showed randomly oriented fibrous mats and spindle-like defects, particularly at high PS-80 concentrations and with low molecular weight PVA (Mw~67 kDa). This is because surfactants can disrupt the balance between surface tension and viscoelastic forces, leading to the formation of beads along the fibers. Therefore, the addition of any kind of surfactants should be carefully controlled during the electrospinning process to avoid unnecessary negative effects on the fiber formation. The average fiber diameters of electrospun PVA/PO-407/PS-80 samples and the fiber diameter distributions are presented in Appendix A and Appendix A, respectively. Images taken for samples prepared from the neat PVA and PVA/PO-407 revealed bead-free, randomly oriented fiber deposition with no gel droplets (Appendix A), while the fiber prepared from the PVA/PO-407 blend showed spindle-like defects (Appendix A). Compared to neat PVA samples, a slight reduction in fiber diameters was observed with the samples containing PO-407 (Appendix A). The reduction in fiber diameters can be interpreted by the lowered surface tension because of the surface-active effect of PO-407. The distribution curves of neat PVA and PVA/PO-407 compositions ranging from normal, skewed, to indefinite shapes were also observed with other combinations (Appendix A). In conclusion, the results indicate that not only polymer grades or concentrations should be considered during fiber preparation but other electrospinning parameters should be adjusted as well. These results confirm that electrospinning depends on multiple factors rather than individual parameters. Choosing the appropriate polymer concentration is critical regarding the quality of fibers, but it does not necessarily mean the formation of fibers of a normal distribution curve. Smooth nanofibrous mats comprising PVA/PO-407 blends have been successfully developed and can be utilized as promising candidates for different pharmaceutical applications, including ophthalmic and thermoreversible systems. These factors might affect the development of nanofiber-based ocular inserts, particularly the in vitro drug dissolution and release kinetics [62]. Since all samples were found to be fibrous, the median concentration of each PVA grade was chosen for further studies.

#### 3.1.2. Polyvinyl Alcohol (PVA)/Tetraethoxysilane (TEOS) Samples

The morphological features of the electrospun samples prepared from the different compositions of TEOS/PVA precursor solutions were studied using SEM (Appendix A). The fiber formation ability of the precursor solutions of different compositions and the morphology of the electrospun sample showed wide variability. For the same polymer ratio, increasing PVA concentration favored the formation of a fibrous structure, whereas increasing TEOS concentration and beady fibrous structures were observed. In the case of S5 and S9 solutions, the interaction between PVA and TEOS leads to the formation of a crosslinked network through the sol–gel process, significantly increasing the viscosity and preventing the solution from flowing through the needle, thus inhibiting fiber formation. There were significant differences in average fiber diameters. At the same PVA/TEOS ratio, the fiber diameter increased with increasing polymer concentration (Appendix A). There is no monotonic trend with increasing TEOS ratios for the same polymer concentration. The highest values are always obtained for the TEOS/PVA 3:2 mass ratio. The fiber diameter distributions of the electrospun fibrous samples are shown in Appendix A.

#### 3.1.3. Cysteamine (Cys)-Loaded Fibers

A SEM was used to study the morphological features and the images of the Cys-loaded electrospun samples of both PVA/PO-407-based and TEOS/PVA-based polymer systems. Images showed randomly oriented fibers with no remarkable beads or gel droplets on the fiber’s surfaces. Regarding the PVA/PO-407-based samples, the images of fibers and the corresponding histogram diameter distribution are displayed in Figure 1. Compared to PVA/PO-407 neat fibers, formulations containing CysH also showed good fiber morphology with spindle-like defects. The addition of CysH to the PVA/PO-407 slightly lowered the average fiber diameter, and the effect was concentration-dependent; formulations loaded with 1.1% (*w*/*w*) (F4, F8, and F12) showed lower fiber diameter compared to formulations with 0.55% (*w*/*w*) (F2 The, F6 and F10). This effect might be attributed to the surface-modifying effect of CysH, which decreases the surface tension and affects the jet flow. The average diameter of fibers for Cys-loaded formulations is summarized in Table 3. The addition of 0.5 to 1% (*w*/*w*) PS-80 improved the morphology of the fibers and eliminated the formation of beads. It has also been observed that the presence of EDTA in the polymeric matrix improved the appearance of fibers. This phenomenon could be attributed to the chelating effect of EDTA. EDTA chelates metal ions, which reduces the free salt concentration and lowers the solution’s conductivity. This change might affect the electrostatic environment during the electrospinning process, thereby influencing the morphology of the resulting fibers (Appendix A). Considering TEOS/PVA-based samples, fibrous bead-free structures were clearly formed regardless of the TEOS/PVA ratio of the precursor solutions used for fiber formation. Images of fibers and the corresponding histogram diameter distribution for the formulations provided the best morphology are displayed in Figure 2. It has been noticed that the addition of the active substance (CysH) reduced beads and improved the fiber formation ability when compared to the neat fibers, where only spindle-like defects were observed. Moreover, the higher the CysH concentration (1.1% (*w*/*w*)), the better fiber the morphology obtained (Appendix A). Therefore, formulations FT9, FT18, and FT27 were chosen for further studies.

The average diameter of fibers for Cys-loaded TEOS/PVA-based formulations is summarized in Table 3. It has been observed that the fiber diameter increases with increasing PVA molecular weight. The effect might be attributed to the increases in the solution viscosity because of increasing the concentration of the solution.

### 3.2. Effect of Different Polyvinyl Alcohol (PVA) Grades, Poloxamer 407 (PO-407), and Polysorbate 80 (PS-80) on Diameter Distribution

According to morphological results, by looking at the average fiber diameters, histograms, values of skewness, and kurtosis, it seems that the formulations are greatly diverse. Samples prepared solely from PVA of different grades favor fiber formation, and the calculated skewness values, reflect normal fiber distribution (coefficients of skewness range from 0.090873 to 0.472034). Except for the skewness value for the PVA sample prepared from low molecular weight and the lowest concentration (P1), the skewness value was between +0.5 and +1 (0.648329), which reflects a moderately skewed distribution (Appendix A and Appendix A).

For the samples prepared from PVA/PO-407, although the addition of PO-407 to the PVA improved the spinnability of the precursor solution, noticeable changes were observed in the fiber distributions of all PVA/PO-407 ratios (Appendix A). This effect can be explained by the fact that the addition of insufficient or too high a surfactant concentration can result in a noncontinuous or improper jet flow, reflected by compound distribution rather than homogenous normal fiber distribution [64,65]. Despite the diversity of fiber distribution, most formulations containing PO-407 have skewness values within an acceptable range from a mathematical point of view. The distribution curves are approximately symmetric (the absolute skewness values were between −0.5 and +0.5). The exception is PP1, which showed substantial skewness (skewness value greater than +1); PP3 and PP6 showed moderate skewness (values were between +0.5 and +1) (Appendix A).

The addition of PS-80 (0.5–1% (*w*/*w*)) to the PVA/PO-407 blend further resulted in different fiber distributions. The distribution curves and their fitting are displayed in Appendix A. The presence of two surfactants in the formulation improved the spinnability of the solution. Nevertheless, the lower concentration of PS-80 (0.5% (*w*/*w*)) resulted in better morphology than the higher concentration (1% (*w*/*w*)). The increased surfactant concentration may lead to an increase in the solution conductivity beyond the critical value, which will consequently interfere with jet formation and fiber deposition [63]. The absolute values of skewness and kurtosis indicate symmetric to moderately skewed distribution curves (the absolute skewness values were between −0.5 and +0.5), while kurtosis values indicate normal to sharp curves (Appendix A).

Most of the formulations showed kurtosis values between +1 and −1, which means most of these curves were not too sharp or too flat, with few exceptions. Some formulations showed good fitting (R^2^ ≥ 0.95) to the Gaussian (normal) distribution, while the rest did not fit [66]. These mathematical results confirmed that electrospinning is a multifactorial process, and to obtain a good fibrous sample, all parameters must be considered. In addition, the morphological characterization of electrospun samples is not only about fiber diameter; instead, further analysis is needed to differentiate between normal and compound distribution curves [64,67]. The particle size distributions have different impacts on the drug release, and some drug delivery systems are intentionally fabricated in the form of multiunit particulate systems to modulate and tailor the drug release from immediate to controlled or modified release pattern. Therefore, from a pharmaceutical point of view, the fibers’ distributions might also have a great impact on the release rate and pattern from electrospun nanofibers prepared for different pharmaceutical applications. The drug formulators should not rely only on the average fiber diameters, but diameter distributions and their effect on the release should be considered. A related work has been published demonstrating the effect of different size distributions on the release of the drug. A study demonstrated the impact of size distribution on the diffusional drug release from numerous particle geometrics (spheres, fibers, and membranes). The results revealed that the size distribution affected the release profiles of spherical particles, followed by fibers, and had no effect on the release from the membrane [68]. This study confirms the concept of considering fiber distribution in the formulation of nanofibers for pharmaceutical applications, particularly when the release is substantially important.

### 3.3. Physicochemical Characterization

Fourier transform infrared spectroscopy (FTIR) was used to study the solid-state characteristics of the electrospun Cys-loaded nanofibers (Figure 3). The spectra suggested the formation of amorphous solid dispersion in both cases (PVA/PO-407 and TEOS/PVA polymeric blends). This can be explained by the formation of new hydrogen bonds along with the reduction of sharp crystalline peaks of CysH. In the case of PVA/PO-407-based samples (Figure 3I), the spectra of the mixtures displayed the presence of the main distinctive peaks of CysH, PVA, PO-407, EDTA, and PS-80. An overlapped absorption peak that appeared at 3296 cm^−1^–3400 cm^−1^ is related to O–H stretching from PVA and the amino group of CysH and EDTA. The overlapped band at 2900 cm^−1^–2920 cm^−1^ is due to C–H stretching from CysH, PVA, and PO-407. The presence of the thiol group (S-H) of CysH is confirmed by the appearance of an absorption band at 2550 cm^−1^–2670 cm^−1^. The absorption band at 1700 cm^−1^–1730 cm^−1^ is attributed to carbonyl (C=O stretching of PVA, PS-80 and EDTA); the band at 1360 cm^−1^–1370 cm^−1^ indicates overlapped CH2 bending stretching of PVA and O-H bending of PO-407; an overlapped peak at 1081 cm^−1^–1100 cm^−1^ is due to C–O stretching from PVA, PO-407 and PS-80; and C–C stretching vibration related to PVA at 828 cm^−1^–840 cm^−1^.

Considering TEOS/PVA-based samples (Figure 3II), the silica shows characteristic peaks between 1100 and 500 cm^−1^, which are related to the asymmetric and symmetric stretching and bending vibrations of Si–O–Si bonds. For the fibrous samples, two characteristic PVA peaks appear at 1700 cm^−1^, which are specific to the C=O groups. These peaks cannot be observed in the spectrum of starting materials. The intensity of these peaks is proportional to the PVA content of the hybrid fibers [57]. The spectra clearly show an increase in cross-linking as the TEOS ratio increases.

The spectra of the electrospun nanofibers of the neat PVAs and PVA/PO-407/PS-80 blend showed the most important peaks for the nanofiber and individual components including different PVA grades, PO-407 and PS-80 (Appendix A). FTIR analysis was conducted to detect any structural changes between the components used for nanofiber fabrication. The spectra of the nanofiber blend (PVA/PO-407/PS-80) showed the presence of the following: an overlapped absorption peak at 2900 cm^−1^–2920 cm^−1^ resulting from PVA and PO-407; an absorption band at 1700 cm^−1^–1730 cm^−1^ related to carbonyl (C=O stretching of PVA and PS-80); an absorption peak at 1360 cm^−1^–1370 cm^−1^ due to overlapped CH2 bending stretching from PVA and O–H bending from PO-407; a peak at 1081 cm^−1^–1100 cm^−1^ due to C–O stretching from (PVA, PO-407, and PS-80); and a peak at 828 cm^−1^–840 cm^−1^ is related to the C–C stretching vibration of PVA [69,70,71,72,73,74,75].

Results obtained from EDAX mapping revealed the presence of individual elements for Cys-loaded nanofibers (PVA/PO-407- and TEOS/PVA-based formulations). Based on the EDAX spectra, the atomic composition of the samples and the distribution of elements within the samples were successfully obtained (Appendix A).

The solid state of Cys-loaded nanofibers was also studied by Raman spectroscopy. The peaks of the main components, such as PVA, PO-407, and TEOS, appeared on spectra measured from the fibers as displayed in Appendix A. The peaks of amorphous Cys are also visible in both formulations (PVA/PO-407- and TEOS/PVA-based nanofibers) as displayed in Appendix A, respectively. However, in case of TEOS/PVA-based nanofibers, the Cys signal was more intense than that of PVA/PO-407-based nanofibers where the content of Cys is higher in TEOS/PVA-based nanofibers (Appendix A). The two characteristic peaks for cysteamine (660 cm^−1^) and PVA (1441 cm^−1^) showed the homogeneous nature of the fibers.

### 3.4. Determination of the Drug Content

All formulations of Cys-loading showed a drug content of 100 (% (*w*/*w*)). The results indicate a homogenous distribution of the Cys all over the polymeric matrix during solution preparation and during electrospinning, because the Cys is freely soluble in water, and the selected polymer blend (PVA/PO-407) is hydrophilic.

### 3.5. In Vitro Drug Release and Release Kinetics

The in vitro releases of Cys based on PVA/PO-407 and TEOS/PVA are displayed in Figure 4. The release of the Cys was rapid and complete (in less than 10 min) in most formulations, regardless of whether the matrices were soluble (PVA/PO-407) or insoluble (TEOS/PVA), and this might be attributed to the free solubility of Cys in water. In the case of PVA/PO-407, the surface-active effects of PO-407 and PS-80 create a synergistic effect in addition to amorphization of the drug by the formation of a nanofiber web. In the case of the TEOS/PVA fibers, the large surface area of the produced nanofibers might result in rapid wetting of the matrix, which renders the environment more favorable for Cys dissolution.

Weibull distribution was used to study the Cys release kinetics from the two polymeric blends, since it can be applied to different release behaviors. The release parameters, including M_∞_, β, τ_d_, and R^2^, are summarized in Table 4. All formulations from the various bases were successfully fitted to the Weibull model (R^2^ > 0.9), indicating linear regression. In the case of the TEOS/PVA system, the values were 0.75 < β < 1, indicating a Case II transport, which is a specific mechanism of drug release from polymeric systems, characterized by non-Fickian diffusion where the rate of drug release is predominantly controlled by the relaxation of the polymer matrix rather than by the concentration gradient of the drug. In this transport, the polymer undergoes significant swelling or glass transition, leading to a sharp front of drug release that moves through the matrix. For instance, it could represent both diffusion and swelling-controlled release, which seems logical since the release occurred from the insoluble matrix. For the PVA/PO-407 matrix, most values were <0.75, indicating Fickian diffusion, while the rest (F8 and F10) were 0.75 < β < 1, indicating a Case II transport, which is a specific mechanism of drug release from polymeric systems. These findings confirm that the model-independent Weibull distribution shape parameter effectively characterizes the underlying structure of the kinetics. The fibrous systems facilitate the combination of diverse matrices within a single system, enabling precise adjustment of the system’s functionality-related properties. The marketed product Cystadrops^®^ efficacy and safety has been studied in two clinical trials. The efficacy of Cystadrops^®^ was assessed in vivo by tracking the cystine crystals and photophobia. The first single-arm clinical study was conducted in eight children and one adult. There was a 30% decrease in the cystine crystal at a median frequency of four daily instillations. The second randomized controlled phase III clinical trial was conducted on 32 patients for 90 days following four drops/eye/day. There was a mean decrease of cystine crystals by 40%. In summary, 0.55% *w*/*v* of cost drops was effective and safe [14,76]. Even though Cystadrops^®^ is a highly viscous formulation that might prolong the release of Cys compared to our formulation, an in vivo human pharmacokinetic study also confirmed that immediate-release Cys is effective in depleting cystine levels, with no statistically significant difference in effectiveness compared to delayed-release formulations, and the burst release of our formulation is important in creating a favorable concentration gradient for the permeability of Cys across the corneal layers [77].

### 3.6. Cytocompatibility Study

A Hen’s Egg Test on Chorioallantoic Membrane (HET-CAM) was conducted to study the cytocompatibility of the nanofiber components (Figure 5). The chorioallantoic membrane (CAM) of the avian embryo is a transparent and highly vascularized membrane that contains arteries, veins, and a complex capillary network. The applied samples, including Cys-loaded nanofibers of different compositions, the empty polymeric nanofibers, and the negative control (Phosphate buffer), resulted in no noticeable redness, coagulation, or bleeding, which indicates the tolerability and cytocompatibility of the two formulation components and their plain fibers when compared to a positive control (0.1 N NaOH). The latter induced a strong hemorrhage on the surface of a chick CAM at embryonic day 9.

### 3.7. Accelerated Stability Study

The SEM was used to track the changes in the morphological status of the freshly prepared Cys-loaded nanofiber and samples stored under stressful conditions (Figure 6). The results suggest stable formulations of both PVA/PO-407 as well as TEOS/PVA polymeric blends. Both formulations F12 and FT27 maintained their fibrous morphology over the period of study (4 weeks). Fourier transform infrared spectroscopy was used to follow the changes in physicochemical characteristics of the freshly prepared Cys-loaded nanofiber and nanofibers exposed to stressful conditions (elevated temperature and humidity) (Figure 7). There were no changes in the functional groups of the two samples (F12 and FT27). All the essential groups have been observed with no marked changes. These results suggest that it is possible to formulate preservative-free, stable solid Cys products that can be used to overcome the stability problem of liquid formulations (Appendix A).

## 4. Conclusions

Two formulation approaches have been investigated for delivering cystine-depleting agents to the eyes of cystinosis patients. Fast-release Cys-loaded nanofibrous OIs have been successfully developed using PVA/PO-407 and TEOS/PVA polymeric blends in different proportions. The morphological studies revealed randomly oriented fibrous structures of nanometer size range. Although there was no great difference within average fiber diameters, different distribution curves were obtained by varying the composition of the precursor solution. The physicochemical properties and supramolecular structure showed good compatibility between the corresponding constituents as detected by FTIR spectra. The release of Cys from the TEOS/PVA system followed Case II transport and non-Fickian diffusion, whereas, in the case of the PVA/PO-407 matrix, most formulations followed Fickian diffusion, while the rest were of non-Fickian diffusion. The stability results of the two formulations showed no major change in morphology or physicochemical characteristics. The developed formulations have good cytocompatibility and show no signs of irritation, redness, or coagulation, as demonstrated by the CAM test. The PVA/PO-407 formulation has the potential to enhance the stability and extend the shelf life of the product. However, it may have limited capacity to sustain the release of the cystine-depleting agent, potentially necessitating frequent administration. Nevertheless, the components of this formulation are well-tolerated and commonly used in ocular formulations. The TEOS/PVA formulation also offers the potential to improve stability and increase shelf life; the production process for cystine-depleting agent/TEOS/PVA formulations might be more complex compared to other matrices. Additionally, there is a need for more clinical data to evaluate the efficacy and safety of this formulation in the management of ocular cystinosis. Both formulation approaches have their advantages and disadvantages, and further research is required to determine the most suitable combination for ocular delivery in cystinosis patients. In light of the results above, it can be concluded that both matrixes could be utilized as a potential alternative for conventional Cys eye formulations with good stability, preservative-free formulation of good cytocompatibility to the eye, and a more convenient way of application without facing premature drainage from the eye sac. By controlling the precursor solution composition of electrospun nanofibers, myriads of nanofiber matrices for different pharmaceutical applications can be obtained with a simple, versatile electrospinning technique.

## 5. Further Perspectives

Ocular cystinosis, a rare genetic disorder, manifests severe symptoms that may result in conditions such as glaucoma, vision impairment, and potential blindness. The current treatment approaches encounter challenges due to the intricate nature of the eye and the inefficiency of orally administered Cys in reaching the cornea effectively. Consequently, topical applications are essential for managing ocular cystinosis. While topical therapies have demonstrated efficacy in regulating cystine crystal levels across different eye layers, Cys eye drops face issues like instability and the requirement for frequent dosing to maintain therapeutic concentrations.

### 5.1. Advantages of Nanofibers as Drug Carriers

Utilizing nanofibers as drug carriers presents several benefits, providing targeted drug delivery as well as sustained and controlled release of drugs. Nanofibers improve bioavailability by enhancing solubility and stability, prolonging the contact time, and improving penetration into deeper eye tissues. They also provide the flexibility of encapsulating a wide range of drugs, including thermolabile substances such as proteins. The development of ocular inserts based on nanofibers shows potential in enhancing drug delivery for treating ocular cystinosis. This novel approach addresses stability challenges and diminishes adverse reactions linked to traditional Cys solutions. Overall, nanofibers as drug carriers offer a combination of properties that make them highly suitable candidates for various drug delivery applications.

### 5.2. Versatility and Additional Benefits

Furthermore, this technology extends beyond cystinosis treatment alone; it can function as a versatile carrier for other orphan drugs or active pharmaceutical agents with similar characteristics. The nanofiber-based ophthalmic inserts also offer supplementary lubrication to the eye, potentially reducing irritation. These inserts are made from biodegradable polymers commonly used in artificial tear eye drops, enhancing their biocompatibility and ease of use.

## Figures and Tables

**Figure 1 pharmaceutics-16-01052-f001:**
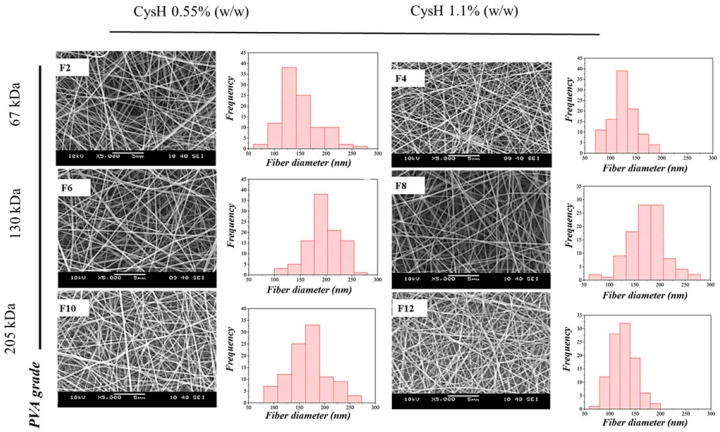
Scanning electron microscope (SEM) images and corresponding histograms of cysteamine-loaded (Cys-loaded) electrospun samples prepared from polyvinyl alcohol (PVA)/PO-407 blends, ethylenediaminetetraacetic acid (EDTA) 0.05% (*w*/*w*), polysorbate 80 (PS-80) (0.5% (*w*/*w*)); cysteamine hydrochloride (CysH) 0.55% (*w*/*w*) (F2, F6, and F10); and CysH 1.1% (*w*/*w*) (F4, F8, and F12) (magnification: 5000×).

**Figure 2 pharmaceutics-16-01052-f002:**
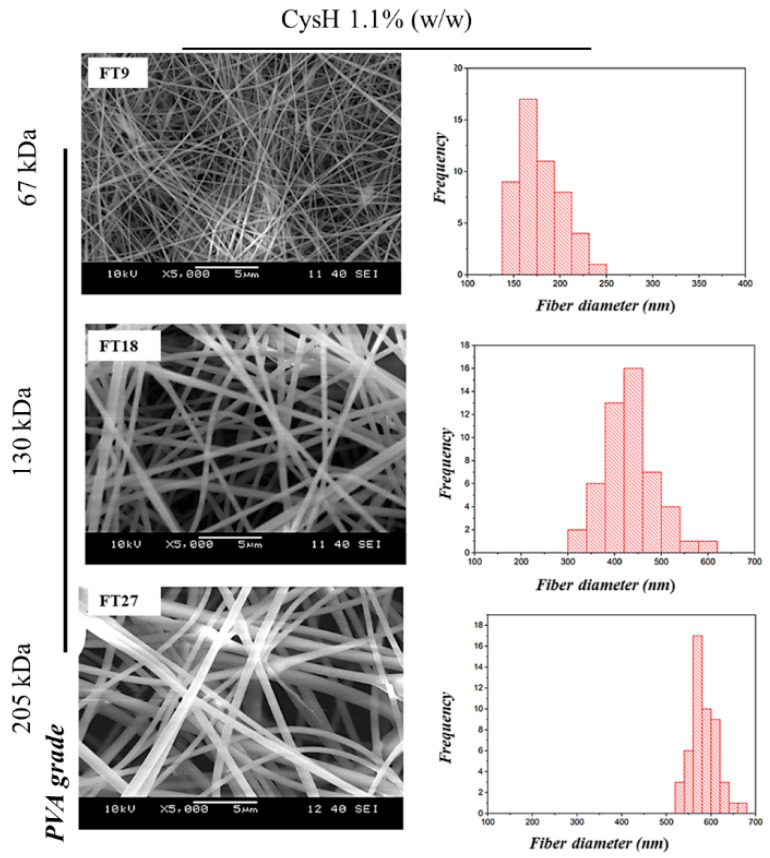
Scanning electron microscope (SEM) images and corresponding histograms of cysteamine-loaded (Cys-loaded) electrospun samples prepared from tetraethoxysilane (TEOS)/polyvinyl alcohol (PVA) and cysteamine hydrochloride (CysH) 1.1% (*w*/*w*). Where FT9, FT18, and FT27 were prepared from Mw~67 kDa, Mw~130 kDa, and Mw~205 kDa of PVA, respectively (magnification: 5000×).

**Figure 3 pharmaceutics-16-01052-f003:**
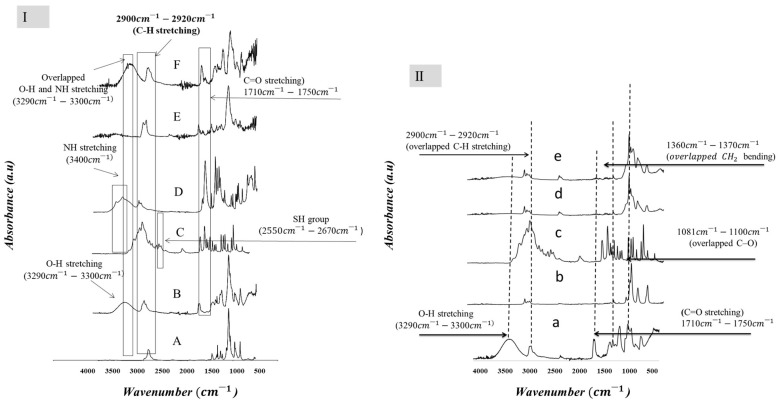
Fourier transform infrared (FTIR) spectra of (**I**) cysteamine (Cys)-loaded samples based on polyvinyl alcohol (PVA)/poloxamer 407 (PO-407) and (**II**) Cys-loaded samples based on tetraethoxysilane (TEOS)/PVA. Where (A): PO-407; (B): PVA (Mw~130 kDa); (C): CysH; (D): Ethylenediaminetetraacetic acid (EDTA, 0.05%); (E): Polysorbate 80 (PS-80); (F): Mixture of the former component. For II, (a): PVA (Mw~130 kDa); (b): TEOS; (c): CysH; (d): TEOS/PVA; and (e): CysH/TEOS/PVA.

**Figure 4 pharmaceutics-16-01052-f004:**
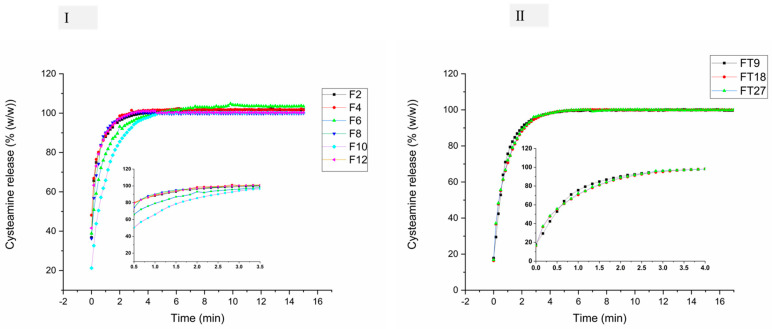
In vitro dissolution profiles of cysteamine HCl (CysH) loaded in (**I**): polyvinyl alcohol (PVA)/poloxamer 407 (PO-407) nanofibers using low molecular weight (PVA, Mw~67 kDa) (F2 and F4); intermediate molecular weight (PVA, Mw~130 kDa) (F6 and F8); and high molecular weight (PVA, Mw~205 kDa) (F10 and F12) and (**II**): tetraethoxysilane (TEOS)/PVA (Mw~130 kDa) (4:1 mass ratio). The dissolution was conducted in phosphate-buffered solution (pH 7.4) at 35 ± 0.5 °C.

**Figure 5 pharmaceutics-16-01052-f005:**
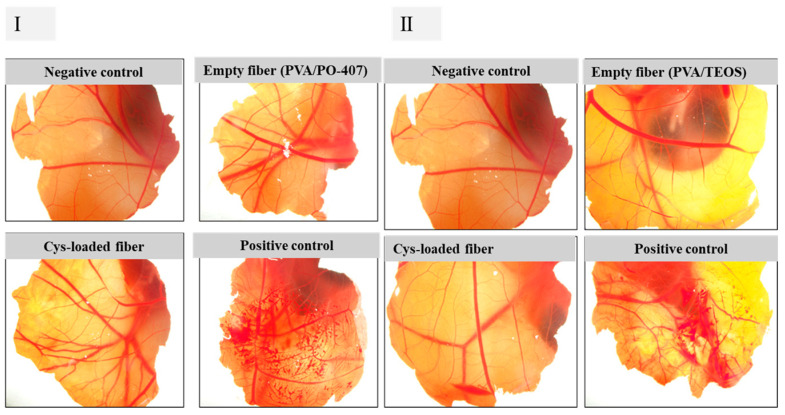
Representative images of Hen’s Egg Test on Chorioallantoic Membrane (HET-CAM). The test was conducted on the vascular structure of the CAM on day 9 of chicken development for a period of 5 min for electrospun cysteamine-loading (Cys-loaded) based on (**I**): polyvinyl alcohol (PVA)/poloxamer 407 (PO-407) and (**II**): tetraethoxysilane (TEOS)/PVA using phosphate-buffered saline (PBS) as a negative control and 0.1 N NaOH as a positive control. A strong hemorrhage induced by 0.1 N NaOH placed on the surface of a chick CAM at embryonic day 9.

**Figure 6 pharmaceutics-16-01052-f006:**
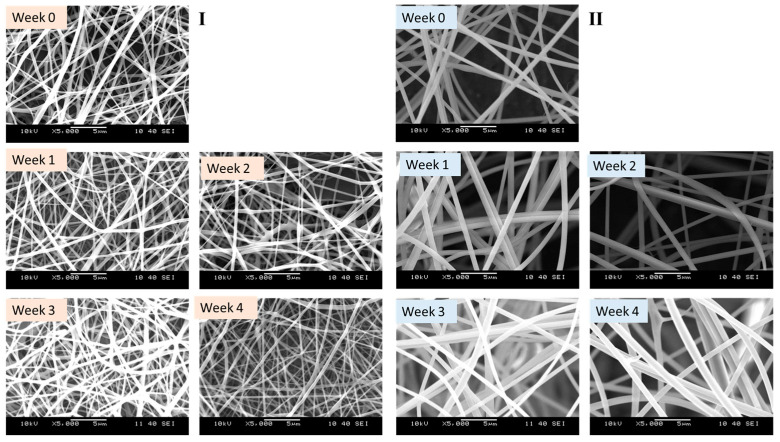
Scanning electron microscopic (SEM) images of the cysteamine (Cys)-loaded fibers at 0, 1, 2, 3, and 4 weeks stored under stress conditions (40 ± 2 °C, 75 ± 5% relative humidity). (**I**): polyvinyl alcohol (PVA)/poloxamer 407 (PO-407) and (**II**) tetraethoxysilane (TEOS)/PVA polymeric blends. (magnification: 5000×).

**Figure 7 pharmaceutics-16-01052-f007:**
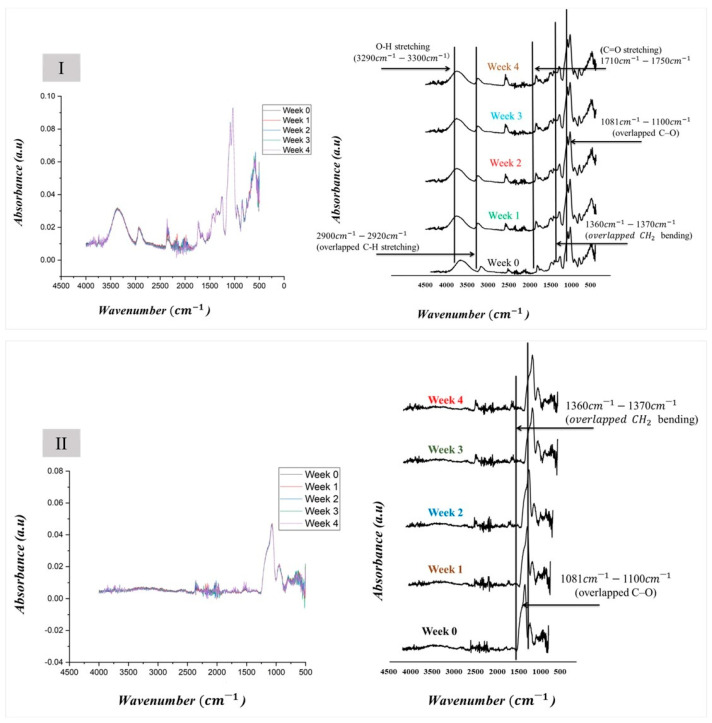
Fourier transform infrared (FTIR) spectra of the cysteamine (Cys)-loaded fibers at 0, 1, 2, 3, and 4 weeks stored under stress conditions (40 ± 2 °C, 75 ± 5% relative humidity). (**I**): polyvinyl alcohol (PVA)/poloxamer 407 (PO-407) and (**II**) tetraethoxysilane (TEOS)/PVA polymeric blends. (In the I part of the figure, the different colors indicate the different storage weeks: black week 0, gold week 1, blue week 2, green week 3, and red week 4.).

**Table 1 pharmaceutics-16-01052-t001:** Composition and the respective amount of cysteamine hydrochloride (CysH)/polyvinyl alcohol (PVA)/poloxamer 407(PO-407)/ethylenediaminetetraacetic acid (EDTA)/polysorbate 80 (PS-80) samples.

PVA/PO-407 (80:20) (Mass Ratio)	CysH Concentration % (*w*/*w*)	Formulation Code	
		CysH/PVA/PO-407	CysH/PVA/PO-407/EDTA/PS-80
Mowiol^®^ 8–88 (17.5% (*w/w*))	0.55	F1	F2
	1.1	F3	F4
Mowiol^®^ 18–88 (14% (*w/w*))	0.55	F5	F6
	1.1	F7	F8
Mowiol^®^ 40–88 (10% (*w/w*))	0.55	F9	F10
	1.1	F11	F12

**Table 2 pharmaceutics-16-01052-t002:** Composition and respective amount of cysteamine hydrochloride (CysH)/polyvinyl alcohol (PVA)/tetraethoxysilane (TEOS) samples.

Polymer Composition	Formulation Code				
PVA/TEOS 1:4 (mass ratio)	PVA-Grade	PVA concentration (% (*w*/*w*))	CysH concentration (% (*w/w*))		
			0	0.55	1.1
	Mw~67 kDa	15	FT1	FT2	FT3
		17.5	FT4	FT5	FT6
		20	FT7	FT8	FT9
	Mw~130 kDa	12	FT10	FT11	FT12
		14	FT13	FT14	FT15
		16	FT16	FT17	FT18
	Mw~205 kDa	5	FT19	FT20	FT21
		7.5	FT22	FT23	FT24
		10	FT25	FT26	FT27

**Table 3 pharmaceutics-16-01052-t003:** Average fiber diameters, skewness, and kurtosis of cysteamine (Cys)-loaded electrospun samples based on polyvinyl alcohol (PVA)/poloxamer 407 (PO-407)/polysorbate 80 (PS-80) polymeric blend, and tetraethoxysilane (TEOS)/PVA polymeric blend.

Sample Code	Morphology	Average Fiber Diameter (nm) ± SD (nm)	Skewness	Kurtosis	Fiber Diameter Distribution (Mono/Polydisperse)	Stability of the Morphology
F2	Fibrous	148 ± 35	−0.15601	0.471756	Polydisperse	Stable
F4	Fibrous	127 ± 25	−0.83847	−0.27836	Polydisperse	Stable
F6	Fibrous	197 ± 35	−1.20192	0.152134	Polydisperse	Stable
F8	Fibrous	175 ± 37	−0.88922	0.455391	Polydisperse	Stable
F10	Fibrous	167 ± 41	−0.22569	0.508247	Polydisperse	Stable
F12	Fibrous	126 ± 24	−1.00682	−0.07583	Polydisperse	Stable
FT9	Fibrous	178 ± 25	−1.70378	−0.16512	Polydisperse	Stable
FT18	Fibrous	434 ± 56	−2.06195	0.600061	Polydisperse	Stable
FT27	Fibrous	584 ± 29	−5.90666	0.211381	Polydisperse	Stable

**Table 4 pharmaceutics-16-01052-t004:** Dissolution kinetic parameters of Cysteamine (Cys) loaded in polyvinyl alcohol (PVA)/poloxamer 407 (PO-407) and tetraethoxysilane (TEOS)/PVA.

Formulation Code	*M_∞_*	*β* Parameter	*τ* * _d_ *	Correlation Coefficient (*R*^2^)
F2	102.05821	0.4548	12.37	0.99787
F4	101.99347	0.58428	18.477	0.99395
F6	103.81088	0.62009	39.872	0.99867
F8	99.89601	0.80627	24.52	0.99769
F10	100.39196	0.97509	69.25	0.999
F12	100.49166	0.63569	18.53	0.99583
FT9	99.79845	0.90072	48.03	0.99785
FT18	100.13645	0.81412	50	0.99821
FT27	100.07967	0.82283	49.212	0.99806

## Data Availability

Data are contained within the article and Appendix A.

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
