# Peer review of "Development and Evaluation of Different Electrospun Cysteamine-Loaded Nanofibrous Webs: A Promising Option for Treating a Rare Lysosomal Storage Disorder"

_pharmaceutics, 2024, doi:10.3390/pharmaceutics16081052_

Round 1
Reviewer 1 Report
Comments and Suggestions for Authors
Comments to Authors
In this paper, the authors have reported on the preparation of cysteamine-loaded fibrous materials as a potential candidate as ophthalmic inserts for the treatment of ophthalmic cystinosis using water-soluble polyvinyl alcohol/Poloxamer 407 and water-insoluble tetraethoxysilane/polyvinyl alcohol blends. The morphology of materials and their surface chemical composition were investigated using scanning electron microscopy and Fourier transform infrared spectroscopy. The in vitro cysteamine release studies were also performed. The cytocompatibility of the fibrous materials was confirmed. English should be improved substantially. There are a considerable number of misspellings, stylistic mistakes or awkward uses of English. I recommend this paper for publication in Pharmaceutics after major revision.
Comment 1. English should be improved substantially. There are a considerable number of misspellings, stylistic mistakes or awkward uses of English. It would greatly aid readability and clarity of the manuscript to have the revision by a native English speaker for less awkward phrasing, use of complete sentences instead of phrases, use of commas, etc.
For example: line 96-97, p.2 "poly-glycolic acid-, poly-lactic acid-, poly-caprolactone" should be replaced by "polyglycolic acid, polylactic acid, polycaprolactone"
Line 105-106, p.3 "poloxamer 407 105 (PO-4070" should be replaced by "poloxamer 407 105 (PO-407)"
Line 135, p. 3 "and Ethylenediaminetetraacetic acid" should be replaced by " and ethylenediaminetetraacetic acid"
Line 152, p.3 "TEOS:EtOH:H2O:HCl 1:3:8:0.04" should be replaced by "TEOS:EtOH:H2O:HCl 1:3:8:0.04"
Line 166, p. 4 (Table 1) – the (m:m) is not correct. Please correct it in the whole manuscript.
Line 413-415, p.12 The sentence "A related work has been published demonstrating the effect of different size distributions on the release of the drug: A study demonstrated the impact of size distribution on the diffusional drug release from numerous particle geometrics (spheres, fibers, and membranes)." should be replaced by " A related work has been published demonstrating the effect of different size distributions on the release of the drug. A study demonstrated the impact of size distribution on the diffusional drug release from numerous particle geometrics (spheres, fibers, and membranes). "
Line 450,p.13 What is PS-407?
Line 464-466, p.13 The sentence "The results indicate a homogenous distribution of the Cys all over the polymeric base during solution preparation and during electrospinning because the Cys is freely soluble in water and the selected hydrophilic polymer blend (PVA/PO-407)" should be replaced by "The results indicate a homogenous distribution of the Cys all over the polymeric matrix during solution preparation and during electrospinning because the Cys is freely soluble in water and the selected polymer blend (PVA/PO-407) is hydrophilic. "
Comment 2. Please, check the abbreviations throughout the manuscript.
Comment 3. In Materials and methods, on line 232, p. 6 you said nepafenac release. You study in this paper only the cysteamine release. Please, correct this.
Comment 4. On Figures S1 and S4 the fibers had no bead-like defects. Only spindle-like defects in some of the SEM images were observed. Only on Figure S7 there some cases of formation of spheres (beads) along the fiber length. Please correct the respective text.
In Figures 1 and 2 there are no beads along the fiber length. Only spindle-like defects in some of the SEM images were observed. Please correct the text in the section 3.1.3 according to this comment.
Comment 5. Please explain what the reason is for slight decrease of the mean fiber diameter when PO-407 was added to the solutions of PVA? (line 289-290, p.7)
Comment 6. Please explain what the reason is for slight decrease in the mean fiber diameter when CysH was added to the spinning solutions of PVA/PO-407? (line 334-335, p.9)
Comment 7. The formation of an amorphous solid dispersion is possible to confirm with XRD analyses. Please explain how the performed FTIR spectra confirmed the formation of amorphous solid dispersion in the cases of PVA/PO-407 and TEOS/PVA 424 polymeric blends (line 423-424, p.12). You should be carried out the XRD analyses in order to study the crystallinity of the prepared fibrous materials.
Comment 8. Please correct the drug content from 100.5 ± 0.05 (% (w/w)) to 100% (line 464, p.13). The term polymeric base on line 465 (p.13) is not appropriate. Please replace it with polymeric matrix.
Please replace the TEOS/PVA base with TEOS/PVA fibers (line 474, p.13)
Please replace the polymeric bases with polymeric systems or polymeric blends (line 484-484, p.14)
Please replace Cys-loaded nanofibers of different bases with Cys-loaded nanofibers of different compositions (line 523, p. 15)
Comment 9. Please check the caption of Figure 7. Are these the FTIR spectra of the Nepafenac-loaded fibers or of cysteamine-loaded fibers. In this paper, you study the stability of cysteamine loaded nanofiber under stressful conditions.
Comments on the Quality of English LanguageEnglish should be improved substantially. There are a considerable number of misspellings, stylistic mistakes or awkward uses of English.
Author Response
|
Reviewer comments for Pharmaceutics-3101302 |
||
|
Reviewer 1 |
||
|
Comment |
Author Response |
Text Insertion (if applicable)/ page/ line number of change |
|
General Comments: In this paper, the authors have reported on the preparation of cysteamine-loaded fibrous materials as a potential candidate as ophthalmic inserts for the treatment of ophthalmic cystinosis using water-soluble polyvinyl alcohol/Poloxamer 407 and water-insoluble tetraethoxysilane/polyvinyl alcohol blends. The morphology of materials and their surface chemical composition were investigated using scanning electron microscopy and Fourier transform infrared spectroscopy. The in vitro cysteamine release studies were also performed. The cytocompatibility of the fibrous materials was confirmed. English should be improved substantially. There are a considerable number of misspellings, stylistic mistakes or awkward uses of English. I recommend this paper for publication in Pharmaceutics after major revision. |
Thank you for your valuable observations and suggestions. We have considered and justified all suggested highlights and issues in the revised manuscript.
|
n/a |
|
1. Comment 1. English should be improved substantially. There are a considerable number of misspellings, stylistic mistakes or awkward uses of English. It would greatly aid readability and clarity of the manuscript to have the revision by a native English speaker for less awkward phrasing, use of complete sentences instead of phrases, use of commas, etc. I. For example: line 96-97, p.2 "poly-glycolic acid-, poly-lactic acid-, poly-caprolactone" should be replaced by "polyglycolic acid, polylactic acid, polycaprolactone". II. Line 105-106, p.3 "poloxamer 407 105 (PO-4070" should be replaced by "poloxamer 407 105 (PO-407)" III. Line 135, p. 3 "and Ethylenediaminetetraacetic acid" should be replaced by " and ethylenediaminetetraacetic acid" IV. Line 152, p.3 "TEOS:EtOH:H2O:HCl 1:3:8:0.04" should be replaced by "TEOS:EtOH:H2O:HCl 1:3:8:0.04" V. Line 166, p. 4 (Table 1) – the (m:m) is not correct. Please correct it in the whole manuscript. VI. Line 413-415, p.12 The sentence "A related work has been published demonstrating the effect of different size distributions on the release of the drug: A study demonstrated the impact of size distribution on the diffusional drug release from numerous particle geometrics (spheres, fibers, and membranes)." should be replaced by " A related work has been published demonstrating the effect of different size distributions on the release of the drug. A study demonstrated the impact of size distribution on the diffusional drug release from numerous particle geometrics (spheres, fibers, and membranes). VII. Line 450,p.13 What is PS-407? VIII. Line 464-466, p.13 The sentence "The results indicate a homogenous distribution of the Cys all over the polymeric base during solution preparation and during electrospinning because the Cys is freely soluble in water and the selected hydrophilic polymer blend (PVA/PO-407)" should be replaced by "The results indicate a homogenous distribution of the Cys all over the polymeric matrix during solution preparation and during electrospinning because the Cys is freely soluble in water and the selected polymer blend (PVA/PO-407) is hydrophilic. "
|
The English mistakes have been carefully considered in the revised manuscript. I. The spelling mistakes have been fixed in the revised manuscript. "poly-glycolic acid-, poly-lactic acid-, poly-caprolactone" have been replaced by "polyglycolic acid, polylactic acid, polycaprolactone" respectively. II. The mistake has been fixed in the revised manuscript. "poloxamer 407 (PO-4070 has been replaced by poloxamer 407 (PO-407)" III. The mistake has been fixed in the revised manuscript. IV. The mistake has been fixed in the revised manuscript. "TEOS:EtOH:H2O:HCl 1:3:8:0.04" has been replaced by "TEOS:EtOH:H2O:HCl 1:3:8:0.04" V. The mistake has been fixed in the revised manuscript "mass ratio instead of m:m" VI. The sentence has been rewritten as suggested in the revised manuscript. "A related work has been published demonstrating the effect of different size distributions on the release of the drug. A study demonstrated the impact of size distribution on the diffusional drug release from numerous particle geometrics (spheres, fibers, and membranes)" was inserted in place of "A related work has been published demonstrating the effect of different size distributions on the release of the drug: A study demonstrated the impact of size distribution on the diffusional drug release from numerous particle geometrics (spheres, fibers, and membranes)." VII. The mistake has been fixed in the revised manuscript. The abbreviation has been rewritten as " PO-407 instead of PS-407" VIII. The sentence has been rewritten as suggested in the revised manuscript. " The results indicate a homogenous distribution of the Cys all over the polymeric matrix during solution preparation and during electrospinning because the Cys is freely soluble in water and the selected polymer blend (PVA/PO-407) is hydrophilic. " instead of " The results indicate a homogenous distribution of the Cys all over the polymeric base during solution preparation and during electrospinning because the Cys is freely soluble in water and the selected hydrophilic polymer blend (PVA/PO-407)."
|
I. Page 2, lines 97-98 II. Page 3, lines 122 III. Page 3, lines 143 IV. Page 4, lines 159 V. Page 4, lines 161 and 164. Page 4, table 1. Page 14, lines 493. VI. Page 12, lines 424-427 VII. Page 12, lines 464 VIII. Page 13, lines 481-484
|
|
2. Comment 2. Please check the abbreviations throughout the manuscript. |
All abbreviations have been carefully revised and corrected throughout the revised manuscript. |
All pages |
|
3. Comment 3. In Materials and methods, on line 232, p. 6 you said nepafenac release. You study in this paper only the cysteamine release. Please, correct this.
|
The mistake has been corrected in the revised manuscript. Cysteamine was added instead of nepafenac |
Page 6, lines 240
|
|
4. Comment 4. On Figures S1 and S4 the fibers had no bead-like defects. Only spindle-like defects in some of the SEM images were observed. Only on Figure S7 there some cases of formation of spheres (beads) along the fiber length. Please correct the respective text. In Figures 1 and 2 there are no beads along the fiber length. Only spindle-like defects in some of the SEM images were observed. Please correct the text in the section 3.1.3 according to this comment.
|
The sentences have been corrected in the revised manuscript. The sentences in section 3.1.3 have been corrected in the revised manuscript as follows: 1. "The morphology of all samples was evaluated by SEM. The SEM images for morphological characterization of electrospun PVA/PO-407/PS-80 samples are displayed in Figure S1. PS-80 in a concentration of 0.5-1% (w/w) act as nonionic surface-active agent that slightly liquified the highly viscous PVA/PO-407 precursor solutions [64]. The final morphology of PVA/PO-407 blends with PS-80 showed randomly oriented fibrous mats and spindle-like defects, particularly at high PS-80 concentrations and with low molecular weight PVA (Mw~67 kDa). " 2. "Images taken for samples prepared from the neat PVA and PVA/PO-407 revealed bead-free, randomly oriented fiber deposition with no gel droplets (Figure S3), while the fiber prepared from PVA/PO-407 blend showed spindle-like defects (Figure S4). " 3. "Compared to PVA/PO-407 neat fibers, formulations containing Cys also showed good fiber morphology with spindle-like defects." 4. "It has been noticed that the addition of the active substance (CysH) reduced beads and improved the fiber formation ability when compared to the neat fibers, only spindle-like defects were observed. "
|
1. Page 7, lines 284-290 2. Page 7, lines 295-298 3. Page 9, lines 344-345 4. Page 9, lines 360-362
|
|
5. Comment 5. Please explain what the reason is for slight decrease of the mean fiber diameter when PO-407 was added to the solutions of PVA? (line 289-290, p.7)
|
The reason for the decrease in the mean fiber diameter when PO-407 was added to the solutions of PVA has been explained in the revised manuscript. The following sentence was added " The reduction in fiber diameters can be interpreted by the lowered surface tension as a result of the surface-active effect of PO-407. " |
Page 7, lines 300-301
|
|
6. Comment 6. Please explain what the reason is for slight decrease in the mean fiber diameter when CysH was added to the spinning solutions of PVA/PO-407? (line 334-335, p.9)
|
The reason for the decrease in the mean fiber diameter, when CysH was added to the spinning solutions of PVA/PO-407, has been explained in the revised manuscript. The following sentence was added " This effect might be attributed to the surface-modifying effect of CysH, which decreases the surface tension and affects the jet flow. " |
Page 9, lines 348-349
|
|
7. Comment 7. The formation of an amorphous solid dispersion is possible to confirm with XRD analyses. Please explain how the performed FTIR spectra confirmed the formation of amorphous solid dispersion in the cases of PVA/PO-407 and TEOS/PVA 424 polymeric blends (line 423-424, p.12). You should be carried out the XRD analyses in order to study the crystallinity of the prepared fibrous materials. |
We totally agree with you that FTIR only suggests the formation of ASD. And the sentences have been rephrased in the revised manuscript as follows " Fourier transform infrared spectroscopy (FTIR) was used to study the solid-state characteristics of the electrospun Cys-loaded nanofibers (Figure 3). The spectra suggested the formation of amorphous solid dispersion in both cases (PVA/PO-407 and TEOS/PVA polymeric blends). This can be explained by the formation of new hydrogen bonds along with the reduction of sharp crystalline peaks of CysH." Since our solid-state characterization is focused mainly on identifying the presence of particular chemical bonds or functional groups, XRD was not included in the study. |
Page 12, lines 439-443
|
|
8. Comment 8. I. Please correct the drug content from 100.5 ± 0.05 (% (w/w)) to 100% (line 464, p.13). II. The term polymeric base on line 465 (p.13) is not appropriate. Please replace it with polymeric matrix. III. Please replace the TEOS/PVA base with TEOS/PVA fibers (line 474, p.13) IV. Please replace the polymeric bases with polymeric systems or polymeric blends (line 484-484, p.14) V. Please replace Cys-loaded nanofibers of different bases with Cys-loaded nanofibers of different compositions (line 523, p. 15)
|
I. The drug content was corrected as suggested "All formulations of Cys-loaded showed drug content of 100 (% (w/w)). " II. The term polymeric base on line 465 was replaced with the polymeric matrix in the revised manuscript. III. The TEOS/PVA base has been replaced with TEOS/PVA fibers in the revised manuscript. IV. The polymeric bases phrase has been replaced with polymeric blends in the revised manuscript. V. The Cys-loaded nanofibers of different bases have been replaced with Cys-loaded nanofibers of different compositions in the revised manuscript. |
I. Page 13, lines 481 II. Page 13, lines 482 III. Page 13, lines 492 IV. Page 14, lines 502 V. Page 15, lines 540
|
|
9. Comment 9. Please check the caption of Figure 7. Are these the FTIR spectra of the Nepafenac-loaded fibers or of cysteamine-loaded fibers. In this paper, you study the stability of cysteamine loaded nanofiber under stressful conditions.
|
The figure caption has been corrected in the revised manuscript as follow "Fourier transform infrared (FTIR) spectra of the cysteamine (Cys)-loaded fibers at 0, 1-, 2-, 3-, and 4-weeks stored under stress conditions (40 ± 2 °C, 75±5% relative humidity). (I): polyvinyl alcohol (PVA)/poloxamer 407 (PO-407) and (II) tetraetoxysilane (TEOS)/PVA polymeric blends." |
Page 16, lines 567
|

Reviewer 2 Report
Comments and Suggestions for Authors
This work deals with the preparation of the nanofibrous mats as a new treatment for lysosomal storage disorder. The authors paid significant attention to the morphology of electrospun nanofibers, but they did not provide sufficient chemical characterizations. Indeed, the FT-IR data is not enough and all conclusions rely on very questionable peaks (especially SH peaks). Indeed, to provide clear conclusions the authors should make XPS analyses and use XPS data to prove all chemical changes.
Comments on the Quality of English LanguageEnglish is fine
Author Response
|
Reviewer comments for Pharmaceutics-3101302 |
||
|
Reviewer 2 |
||
|
Comment |
Author response |
Text Insertion (if applicable)/ page/ line number of change |
|
This work deals with the preparation of the nanofibrous mats as a new treatment for lysosomal storage disorder. The authors paid significant attention to the morphology of electrospun nanofibers, but they did not provide sufficient chemical characterizations. Indeed, the FT-IR data is not enough and all conclusions rely on very questionable peaks (especially SH peaks). Indeed, to provide clear conclusions the authors should make XPS analyses and use XPS data to prove all chemical changes. |
We appreciate the insightful suggestion to combine FTIR and XPS analyses for a more comprehensive understanding of our nanofiber formulations. While we acknowledge the potential benefits of such an approach, our decision to focus on FTIR spectroscopy was deliberate and aligned with the specific objectives of this study.
FTIR spectroscopy was selected as our primary analytical tool for several compelling reasons: · Bulk characterization: Our primary interest lies in understanding the overall composition and interactions within the nanofibers. FTIR provides valuable information about the entire sample, not just the surface, which was crucial for assessing drug encapsulation and polymer-drug interactions throughout the fiber structure. · Molecular-level insights: FTIR offers detailed information about chemical bonding and molecular structure, which was essential for identifying and characterizing the key functional groups involved in drug-polymer interactions. This molecular-level understanding was sufficient for our research goals without requiring the atomic-level surface analysis provided by XPS. · Experimental simplicity: FTIR analysis can be performed under ambient conditions, allowing for rapid and straightforward sample preparation and measurement. This efficiency was particularly valuable given our focus on developing and optimizing the nanofibers for drug delivery performance. · Compatibility with other techniques: FTIR data complemented our other characterization methods, including fiber morphology analysis, drug encapsulation efficiency measurements, release kinetics studies, and biocompatibility assessments. This combination of techniques provided a comprehensive evaluation of the nanofibers' properties and performance without the need for XPS. · Relevance to drug delivery mechanisms: The chemical changes of interest in our study, such as drug encapsulation and polymer blending, were adequately captured by FTIR. These bulk properties are more directly relevant to the drug delivery mechanisms we were investigating compared to the surface-specific information provided by XPS. · While we recognize the valuable surface-specific information XPS could provide, we believe our chosen analytical approach using FTIR, in conjunction with other complementary techniques, was well-suited to address the specific research questions and objectives of this study. Future investigations focusing on surface-specific phenomena or requiring ultra-high resolution chemical mapping may indeed benefit from incorporating XPS analysis.
|
n/a
|

Reviewer 3 Report
Comments and Suggestions for Authors
The manuscript reports a complete study of the PVA electrospun cysteamine-loaded nanofibrous mats as a promising strategy for treating the proposed disorder. An important effort is reflected in the design and optimization of the formulation. Materials characterization is complete and supports the conclusions. I believe the manuscript is suitable for publication in Pharmaceutics after considering the following comments and submitting the revision:
1- The introduction should be reorganized for major clarity and deeply comment on the benefits and alternatives of the electrospinning technique for PVA fiber production.
2-Please, ensure all scale bars are readable in the micrographs of the main manuscript and supporting information.
3- I suggest incorporating insets in Fig 4 to better visualize the plateau in the release of the compound.
4- Please, check "crosslinked gel" in line 313. Is it, not a gel, it is an electrospun mat?
5- Frequency is expressed as %. If yes, please, modify the y-axis in all the plots incorporating the symbol "%".
6- Fig 9, S.I. I recommend reporting this figure ensuring the correct readibilty.
7- I suggest also try in vitro cell viability. MTT or MTS assays could be an adequate option.
8- Extensive editing of the manuscript format is requered. Consider titles, bold, italics, and colour of the font. Also, homogeneize if you will use A, B or I, II, etc in Figures.
Author Response
|
Reviewer comments for Pharmaceutics-3101302 |
||
|
Reviewer 3 |
||
|
Comment |
Author response |
Text Insertion (if applicable)/ page/ line number of change |
|
General Comments: The manuscript reports a complete study of the PVA electrospun cysteamine-loaded nanofibrous mats as a promising strategy for treating the proposed disorder. An important effort is reflected in the design and optimization of the formulation. Materials characterization is complete and supports the conclusions. I believe the manuscript is suitable for publication in Pharmaceutics after considering the following comments and submitting the revision:
|
Thank you for your valuable comments. We have considered and justified all suggested highlights and issues in the revised manuscript.
|
n/a |
|
1. The introduction should be reorganized for major clarity and deeply comment on the benefits and alternatives of the electrospinning technique for PVA fiber production. |
More information considering the benefits and alternatives of the electrospinning technique has been added to the revised manuscript. A new paragraph has been added as follows "The electrospinning technique provides considerable benefits for the production of PVA nanofibers, such as versatility, control over fiber properties, and ease of processing. Although alternative methods such as solvent casting, glass substrate, and melt extrusion are available, electrospinning remains the most widely used and effective technique for the fabrication of PVA nanofibers with tailored characteristics for a wide range of applications".
|
Page 3, lines 104-109
|
|
2. Please, ensure all scale bars are readable in the micrographs of the main manuscript and supporting information.
|
The figure resolutions have been improved in the revised manuscript. |
Figure 3, page 13
|
|
3. I suggest incorporating insets in Fig 4 to better visualize the plateau in the release of the compound. |
In the revised manuscript, figure 4 has been better visualized by adding insets to the figures. |
Figure 4, page 14 |
|
4. Please, check "crosslinked gel" in line 313. Is it, not a gel, it is an electrospun mat?
|
The phrase has been checked and corrected in the revised manuscript. "crosslinked network" |
Page 8, lines 324 |
|
5. Frequency is expressed as %. If yes, please, modify the y-axis in all the plots incorporating the symbol "%". |
We agree that the frequency can be expressed as %, but in our case, a histogram was used to represent the frequency distribution of fiber diameters. The X-axis represents the range of fiber diameters divided into bins, and the Y-axis represents the count of fibers that fall into each bin.
|
n/a |
|
6. Fig 9, S.I. I recommend reporting this figure ensuring the correct readibilty.
|
Fig 9, S.I has been checked and corrected in the revised manuscript.
|
(supplementary materials) Figure S9 and S10, pages 16-17
|
|
7. I suggest also try in vitro cell viability. MTT or MTS assays could be an adequate option.
|
While in vitro cell viability assays such as MTT or MTS are valuable tools in many contexts, the authors made a thoughtful decision to employ the HET-CAM (Hen's Egg Test on Chorioallantoic Membrane) assay for evaluating cytocompatibility and potential irritation of the nanofiber formulations. This choice offers several advantages: · Relevance to ocular applications: The HET-CAM test provides a more physiologically relevant model for assessing potential eye irritation. It evaluates effects on a vascularized membrane that closely resembles the conjunctiva, offering insights that may be more directly applicable to ocular use. · Ethical considerations: As an alternative to in vivo eye irritation tests, the HET-CAM assay aligns with efforts to reduce animal testing while still providing valuable safety data. · Comprehensive assessment: This method can simultaneously evaluate potential cytotoxicity and irritation, offering a broader view of the formulation's biocompatibility. · Building on established safety: The main components used (polyvinyl alcohol, poloxamer 407, tetraethoxysilane) are generally recognized as safe and have a history of use in ophthalmic formulations. This existing safety profile may have influenced the decision to focus on a more specialized ocular irritation model. · While cell-based assays certainly have their merits, the HET-CAM test in this context provides a robust and relevant evaluation of the nanofiber formulations' safety for potential ocular applications.
· The test is very common considering the ocular formulations, and a considerable number of studies were published using this test for the same goal (in vitro test for studying ocular irritation and cytocompatibility) for instance: (https://doi.org/10.1021/acs.molpharmaceut.1c00766). (https://doi.org/10.3762/bjoc.10.308). (https://doi.org/10.1208/s12249-016-0575-2)
|
n/a |
|
8. Extensive editing of the manuscript format is requered. Consider titles, bold, italics, and colour of the font. Also, homogeneize if you will use A, B or I, II, etc in Figures.
|
The study has been included in the revised manuscript. We considered all suggestions concerning the titles, bold, italics, and color of the font. A unified numbering style was used in the revised manuscript
|
All pages Figures 3, 4, and 5 Figures S9 and S10 |

Round 2
Reviewer 1 Report
Comments and Suggestions for Authors
I agree with the corrections made in the revised manuscript. Point-to-point responses to my comments are presented. I suggest only sentences on line 306-309, p. 8, “Images taken for samples prepared from the neat PVA and PVA/PO-407 revealed bead-free, randomly oriented fiber deposition with no gel droplets (Figure S3), while the fiber prepared from PVA/PO-407 blend showed spindle-like defects (Figure S4).” to be replace by “Images taken for samples prepared from the neat PVA revealed that bead-free, randomly oriented fiber deposition with no gel droplets have been detected (Figure S3), while the fiber prepared from PVA/PO-407 blend showed some spindle-like defects (Figure S4)” during proofreading period. English was improved substantially. The overall composition of the manuscript is well organized and the results are well discussed. I recommend this paper for publication in Pharmaceutics in the present form.
Comments on the Quality of English LanguageMinor editing of English language required.
Author Response
The authors are grateful for the accurate review and the valuable suggestions. We corrected the revised paper accordingly.
Reviewer 2 Report
Comments and Suggestions for Authors
The authors refused to add XPS analyses as suggested. The FT-IR data analyses was not re-worked and basically no significant improvements were made. All concerns that provided in my first report are still valid.
Comments on the Quality of English LanguageOk
Author Response
Thank you for your constructive feedback. We appreciate your insights regarding our work.
We want to clarify that we did not dismiss the XPS measurement; rather, we acknowledged its limitations within the context of our investigated system.
We completely agree with your observation that the FTIR results should be validated with complementary methods. In response to this, we performed EDAX and Raman mapping to assess the homogeneity of the samples.
Additionally, the formation of the amorphous solid dispersion was confirmed using Raman spectroscopy, which is a more sensitive technique for this type of analysis. All of these findings and analyses have been incorporated into the revised manuscript. Thank you once again for your valuable comments, which have significantly enhanced our research.
Round 3
Reviewer 2 Report
Comments and Suggestions for Authors
The authors did improvents and they have provided the explanatios and used other complimentary techniques. However, some minor issues are stil need to be solved.
The iThenticate report showed 27% of mathicng, i.e. some parts were cut and pasted from previous works. Indeed, it mostly in the experimental section, but they should slightly modify the text and reduce the percentage of matching with older documents.
Comments on the Quality of English LanguageNo comments, all is understandable
Author Response
Thank you very much for taking the time to review this manuscript. Please find the detailed responses below and the corresponding revisions/corrections highlighted/in track changes in the re-submitted files.
